# Health Characteristics of Patients with Cystic Fibrosis whose Genotype Includes a Variant of the Nucleotide Sequence c.3140-16T>A and Functional Analysis of this Variant

**DOI:** 10.3390/genes12060837

**Published:** 2021-05-28

**Authors:** Elena Kondratyeva, Tatyana Bukharova, Anna Efremova, Yuliya Melyanovskaya, Natalia Bulatenko, Ksenia Davydenko, Alexandra Filatova, Mikhail Skoblov, Stanislav Krasovsky, Nika Petrova, Alexander Polyakov, Tagui Adyan, Elena Amelina, Vera Shadrina, Elena Zhekaite, Aysa Zodbinova, Alexander Chernyak, Rena Zinchenko, Sergei Kutsev, Dmitry Goldshtein

**Affiliations:** 1Research Centre for Medical Genetics, 115522 Moscow, Russia; bukharova-rmt@yandex.ru (T.B.); anna.efremova.83@gmail.com (A.E.); melcat@mail.ru (Y.M.); bnv695@gmail.com (N.B.); xeerox2008@gmail.com (K.D.); maacc@yandex.ru (A.F.); mskoblov@gmail.com (M.S.); sa_krasovsky@mail.ru (S.K.); npetrova63@mail.ru (N.P.); polyakov@med-gen.ru (A.P.); tagui.adyan@yandex.ru (T.A.); eamelina@mail.ru (E.A.); elena_zhekayte@mail.ru (E.Z.); azodbinova@mail.ru (A.Z.); achi2000@mail.ru (A.C.); renazinchenko@mail.ru (R.Z.); kutsev@mail.ru (S.K.); dvgoldshtein@gmail.com (D.G.); 2Faculty of Pediatrics, Perm State Medical University, 614990 Perm, Russia; verashadrina@mail.ru

**Keywords:** c.3140-16T>A variant, CFTR, cystic fibrosis

## Abstract

Cystic fibrosis (CF) is the most common monogenic autosomal recessive disease, associated with pathogenic variants in the CFTR gene. The splicing variant c.3140-16T>A (3272-16T>A) has been described previously and, according to the Russian CF Patients Registry, occurs with a frequency of 0.34%. The phenotypic features of CF patients with the c.3140-16T>A variant were compared with those of patients with the genotype F508del/F508del. Patients with the allele c.3140-16T>A had higher average age and age at diagnosis, and the allele was present in a greater proportion of adults. Patients carrying the c.3140-16T>A allele were characterised by better physical development indicators, both in adults and in children, had preserved pancreatic function, as well as the absence of a number of complications, and required pancreatic enzyme replacement therapy less often than patients with the F508del/F508del genotype. Sweat test values also were lower in patients with the c.3140-16T>A genotype. According to the results of clinical and laboratory studies, the phenotype of patients with the genetic variant c.3140-16T>A can be considered “mild”. Functional CFTR protein activity in the presence of c.3140-16T>A was evaluated using intestinal current measurements (ICM) and the forskolin-induced swelling assay on organoids obtained from patients’ rectal biopsies. c.3140-16T>A had high residual CFTR channel activity and was amenable to effective pharmacological correction with thea VX-770 potentiator. To evaluate the effect of the variant on CFTR pre-mRNA splicing we performed a minigene assay, as well as RT-PCR analysis of RNA isolated from the nasal epithelium and rectal biopsy of patients. We showed that the c.3140-16T>A variant creates a novel acceptor AG dinucleotide within CFTR intron 19, resulting in a 14-nucleotide extension of exon 20. This frameshift produces a premature termination codon and triggers mRNA degradation by the nonsense-mediated decay (NMD) mechanism. Moreover, we observed that the c.3140-16T>A allele could produce a residual amount of normally spliced transcript, thus explaining the patient’s mild phenotype.

## 1. Introduction

Cystic fibrosis (CF) is the most common monogenic autosomal recessive disease (the frequency in Russia is 1:10,000) [1] and is characterised by damage to all exocrine glands of vital organs and systems, as well as respiratory disorders. CF is mainly found in Caucasian individuals and often has an unfavourable prognosis. The disease is caused by a mutation in the CF transmembrane conduction regulator (*CFTR*) gene, which contains 27 exons and is located in region 31.1 of the long arm of chromosome 7 (7q31.1). This gene encodes a protein that is a cAMP-dependent anion channel located on the apical surface of epithelial cells lining the respiratory tract, gastrointestinal tract, pancreas, sweat, bile ducts and part of the reproductive organs. Disruption of the function of the CFTR protein leads to a decrease in the transport of chloride ions and increases the absorption of sodium ions, resulting in a decrease or complete cessation of fluid diffusion through the epithelial membrane [2,3,4]. The main clinical manifestations of the disease in terms of the respiratory tract are associated with viscous bronchial secretions, frequent lung infections, and airway obstruction. In addition to the bronchopulmonary system, the disease causes damage to the pancreas, accompanied by a decrease in the excretion of enzymes [2,3,4].

To date, more than 2000 pathogenic variants in the nucleotide sequence of the *CFTR* gene have been described (http://www.genet.sickkids.on.ca/, accessed on 5 May 2021). As of 31 July 2020 on the CFTR2 website (https://www.cftr2.org/, accessed on 27 May 2021), 442 variants were annotated; 360 of them were pathogenic, 48 were variants with different clinical consequences not related to CF-23, and 11 were variants of unknown significance. In the register of patients with CF of the Russian Federation in 2018, 210 pathogenic variants of *CFTR* were listed, of which 36 pathogenic CFTR mutations occurred with a frequency of >0.1% [5]. Depending on the effect on protein function, all genetic variants have been divided into VII classes. Classes I-III of genetic variants cause a more severe course of the disease, in contrast to classes IV and V, in which the function of the chloride channel is partially preserved. The definition of the corresponding phenotype as ”severe” or “mild” is usually correlated with preservation of the function of the pancreas.

Class VII includes variants that result in the impaired formation of mRNA. These can comprise extensive rearrangements of the *CFTR* gene (deletions, insertions), covering several exons and disrupting the normal structure of the gene and normal splicing (an example is the CFTRdele2,3 deletion common in Russia), or variants that change the donor or acceptor splicing sites of a single exon (for example, 1717-1G>A). Mutations of Class VII are so-called irremediable mutations because they cannot be rescued pharmacologically by themselves and include, for example, large deletions, such as the CFTRdele2, 3 (21 kb) mutation [6].

In 2015, the genetic variant c.3140-16T>A (3272-16T>A) was first described by Krasovsky et al. In a sample of 10 adult CF patients from Russia, its “mild” phenotypic manifestations were demonstrated [7]. In addition, the Russian authors described three clinical observations associated with this variant and similarly emphasised the “mild” course of the disease in patients harbouring this variant [8]. At the same time, this genetic variant encodes a replacement of thymine with adenine in the 19th intron in the 16 position before exon 20. This variant leads to activation of the cryptic site of acceptor splicing in intron 19, which leads to the elongation of exon 20 by 14 nucleotides with a shift in the reading frame and the premature formation of a stop codon, which is not consistent with the phenotypic manifestations of this variant in the *CFTR* gene. However, in the international databases CFTR 1 and CFTR 2, the genetic variant c.3140-16T>A has not been described. The aim of this study was to describe the clinical and laboratory characteristics of the health of patients with CF whose genotype includes the nucleotide sequence c.3140-16T>A and to conduct a functional analysis of this genetic variant.

## 2. Materials and Methods

### 2.1. Materials

To assess the clinical characteristics of the genetic variant c.3140-16T>A, data from the Russian cystic fibrosis patient registry for 2018 were analysed, including information on 3142 patients, 3091 living and 51 deceased [5]. Information from 81 regions from 81 regions is presented. There are currently 85 regions in the Russian Federation. Four regions did not provide data in 2018. We studied the characteristics of a group of patients with the genotype allele c.3140-16T>A (group 1), consisting of 19 people (0.6 males: 0.4 females). The comparison was performed with a group of patients homozygous for the F508del allele (c. 1521_1523delCTT; group 2). The genetic pathogenic variant F508del is the most common in Europe and Russia (allelic frequency according to the Russian Registry for 2018 = 53.1%) and belongs to the class II of genetic variants. Patients who are homozygous for the F508del variant are characterised by a classic clinical course of CF with severe lung and pancreatic damage. The analysis included data from 881 patients with the F508del/F508del genotype, including 707 (80.2%) children and 174 (19.8%) adults. The ratio was 0.5 males to 0.5 females (442:439). The names of the genetic variants are presented according to the traditional nomenclature and according to the coding DNA.

### 2.2. Methods

We used the standard phenol-chloroform extraction method for the isolation of total DNA from the whole blood of patients. PCR and restriction fragment length polymorphism analysis were used for the genotyping of polymorphic gene variants. In the genetic study of the *CFTR* gene mutations, the multiplex amplification technique was used to detect insertion/deletion mutations, and the allele-specific ligation method with subsequent amplification was used to register point mutations. The nucleotide sequences of different patients were determined by direct automatic sequencing on an Applied Biosystems device according to the manufacturer’s protocol. DNA diagnostics were performed according to the consensus algorithm “Cystic fibrosis: definition, diagnostic criteria, therapy” and “genetics of cystic fibrosis: molecular genetic diagnostics of cystic fibrosis” [3]. Genetic research in the Russian Federation includes three stages. Stage 1 involved identification of 35 frequent genetic variants, including c.3140-16T>A. At stage 2, Sanger sequencing was carried out or NGS technologies were used (the region of location of the c.3140-16T>A variant was included in area of study, and the coverage at this point was greater than 10x). DNA analysis of patients was performed on a new generation sequencer (Ion S5). For sample preparation, the technology of ultramultiplex PCR coupled with subsequent sequencing (AmpliSeq™) was used. At the 3rd stage, the MLPA method was used.

The following indicators were taken into account: patient age, age at diagnosis, sweat test score, body mass index (BMI; kg/m^2^), spirometric parameters, including forced expiratory volume in 1 s (FEV1) and forced vital capacity of the lungs (FVC), the colonisation of the bronchopulmonary system by microorganisms (*Pseudomonas aeruginosa*, *Staphylococcus aureus*, *Methicillin-Resistant S*. *aureus (MRSA)*, *Burkholderia cepacia* complex, *Achromobacter* spp., *Stenotrophomonas maltophilia*, nontuberculosis mycobacteria, and gram-negative microflora), pancreatic insufficiency (pancreatic elastase-1 (<200 mcg/g of faeces), complications (meconium sileus, diabetes, osteoporosis, allergic bronchopulmonary aspergillosis, etc.), treatment and therapy. The state of lung function was analysed according to the FVC of the lungs and the FEV1. Studies were conducted in accordance with the ERS/ATS criteria for a group of children who were able to perform respiratory manoeuvres during spirometry [9].

Nutritional status was calculated based on body weight, height, and age, and the Quetelet body mass index (BMI; weight (kg)/height (m^2^)) [10]. BMI percentiles were calculated using the World Health Organization (WHO) program, WHO Anthro (for children under five years of age) and WHO Anthroplus (for children over five years of age) (http://www.who.int/childgrowth/software/en/, accessed on 27 May 2021 and http://www.who.int/growthref/tools/en/, accessed on 27 May 2021). Values from the 26th to the 75th percentile were considered normal. The target value for children and adolescents with CF was the indicator corresponding to normal figures for healthy children of the same sex and age, i.e., the 50th percentile [11]. To assess the growth and weight indicators of young children (up to two years), the mass-growth index (actual weight/ideal weight by height and sex × 100%) was used [10]. For adults with CF, the target BMI values were 22 kg/m^2^ for women and 23 kg/m^2^ for men [11]. The WHO recommends that adolescents and adults be diagnosed with malnutrition if their BMI is <18.5 kg/m^2^ (Report of a WHO Expert Committee, 1995).

### 2.3. RT-PCR Analysis

For analysis of mRNA structure, we used the patient’s (c.3140-16T>A/F508del) and his/her mother’s (carrier of F508del) nasal epithelium cells obtained by brushing nasal cavity with interdental brushes [12], as well as the patient’s intestinal tissue sample obtained by biopsy.

Total RNA was isolated using ExtractRNA reagent (Eurogen, Moscow, Russia) and treated with DNAseI (Thermo Fisher, Waltham, Massachusetts, MA, USA). cDNA was obtained using MMuLV H-reverse transcription system (DIALAT Ltd., Moscow, Russia). PCR was performed with the following primers: CFTR Ex19F (5 ’AGCTATAGCAGTTGTCGCAGT 3′) and CFTR Ex21R (5 ’CCCACTGCAATGTACTCATGA 3′) for detection of splicing changes in the locus containing c.3140-16T>A; CFTR Ex10F (5 ’CTAATGGTGATGACAGCCTCT 3′) and CFTR Ex12R (5 ’AATTCTTGCTCGTTGACCTCC 3′) for the locus containing F508del variant. The PCR-products were analyzed by 2% agarose gel electrophoresis and Sanger sequencing.

#### 2.3.1. Minigene Splicing Assay

To perform the minigene splicing assay we used a pSpl3-Flu2 vector [13].

*CFTR* exon 20 with 300 bp of flanking intronic sequences were amplified from c.3140-16T>A/F508del patient’s genomic DNA with primers: CFTR-Ex20F (5’ GCCAGAAAACTCCCAGTGGT 3’) and CFTR Ex20R-2 (5’ AACAATGGAAATTCAAAGAAATCA 3’). PCR-product was cloned into pSpl3-Flu2 vector by EcoRI/BamHI. The obtained wild-type (WT) and c.3140-16T>A plasmids were verified by Sanger sequencing. 

The minigene plasmids were separately transfected into HEK293T cells by calcium phosphate transfection. After 48 h, cells were harvested and total RNA and cDNA were obtained in the same way as the mRNA analysis. To detect splicing alterations, minigene-specific cDNA was amplified with plasmid-specific primer TurboFP-F (5′ ACAAAGAGACCTACGTCGAGCA 3′) and a *CFTR* gene specific primer MG-CFTR Ex20R (5’ ’ATGGAAATGAAGGTAACAGCAA 3′). The PCR products were analyzed by 12% denaturing urea polyacrylamide gel electrophoresis and Sanger sequencing.

#### 2.3.2. NMD Inhibition

In order to inhibit the nonsense-mediated mRNA decay, the patient’s intestinal organoids culture was treated with 300 μM cycloheximide. After a 6-h incubation, mRNA was isolated and processed as described above. 

#### 2.3.3. Targeted Next Generation Sequencing of PCR-Product

PCR product for NGS was obtained from intestinal organoid cDNA of healthy control and c.3140-16T>A homozygous patient using primers CFTR Ex2F (5′ CGCCTGGAATTGTCAGACATA 3′) and CFTR Ex6R (5′ CATCATTCTCCCTAGCCCAG 3′). NGS libraries were prepared and sequenced on an Ion Torrent S5 (with coverage >200,000). The raw sequencing data were processed with a custom pipeline based on open-source bioinformatics tools HISAT2, Samtools, SAJR. Splice junctions and were visualized using Sashimi plot in IGV.

### 2.4. Methods to Evaluate the Functional Activity of the CFTR Channel

As a sweat test, the conductivity of sweat was determined using the “Nanoduct” system (Wescor, South Logan, Utah, UT, USA). The assessment was performed according to the recommendations of the international consensus [14].

### 2.5. Intestinal Current Measurement Method (ICM)

ICM is based on the determination of the functional activity of ion channels, including chloride channels, based on the epithelial cell membranes of rectal (intestinal) biopsies obtained from the patient in response to the administration of stimulants such as amiloride, forskolin/IBMX, genistein, carbachol, 4,4′-diisothiocyanatostilbene-2,2′-disulfonic acid (DIDS) and histamine. The study was conducted after the patients’ representatives provided signed informed voluntary consent to participate in the study using ICM. The ICM method was conducted according to the European standard operating procedure V2.7_26.10.11 (SOP) [15].

In the first stage, each of the four recirculation chambers was calibrated separately on the VCC MC 8B421 Physiological Instrument (San Diego, CA, USA). Physical factors were considered, such as the presence of air in the contact tips with agar and the resistance of the liquid, as well as environmental factors such as the absence of vibrations near the equipment, accidental contacts with the electrodes and the absence of extraneous working devices in the office. In the second stage, after calibration of the device, the rectal biopsy material was placed in the chamber. Biopsy samples were collected using Olympus Disposable EndoTherapy EndoJaw Biopsy forceps (model #FB-23OU) according to the manufacturer’s instructions. The size of the biopsy sample was 3–5 mm. The biopsy material was placed in a special slider P2407B with a 1.2 mm diameter diaphragm, which was then inserted into the Ussing chambers. The chambers were filled with Meyler buffer solution. The buffer was prepared before the study and included the following: 105 mM NaCl, 4.7 mM KCl, 1.3 mM CaCl2•6H_2_O, 20.2 mM NaHCO_3_, 0.4 mM NaH_2_PO_4_•H_2_O, 0.3 mM Na_2_HPO_4_, 1.0 mM MgCl_2_•6H_2_O, 10 mM HEPES, and 10 mM D-glucose, as well as 0.01 mM indomethacin. The biopsies were heated to 37 °C using a circulation pump connected to a temperature-controlled water bath and continuously carbonated with 95% O_2_ and 5% CO_2_. The registration of the study began with the recording of the basal short-circuit current (preamiloride stage). At the third stage, stimulators (Sigma-Aldrich, Merck) were added in the following sequence: amiloride (100 mM), forskolin (10 mM)/IBMX (100 mM), genistein (100 mM), carbachol (100 mM), DIDS (100 mM) and histamine (100 mM). The study was completed after the basal short-circuit current was recorded. Stimulants were characterised as follows: amiloride inhibits ENaC (sodium channels), forskolin/IBMX (3-isobutyl-1-methylxanthine) activates cAMP-dependent chloride channels (CFTR), genistein activates the opening of the CFTR channel, carbachol initiates the opening of the Ca^2+^ (calcium channel), DIDS is an inhibitor of anionic transport through biological membranes and histamine reactivates the Ca^2+^-dependent secretory pathway.

The results of the Multicenter Intestinal Current Measurements in Rectal Biopsies from CF and non-CF Subjects to Monitor CFTR Function were used as a control for the obtained data, as well as their own comparison groups consisting of ten healthy people without hereditary pathology and diseases of the pancreas and gastrointestinal tract (control group) and three patients with CF with the genotype c.1521_1523delCTT (p.Phe508del, F508del) in the homozygous state. The studies were performed on the intestinal biopsy of one patient with genotype c.3140-16T>A/c.3140-16T>A and one patient with genotype c.3140-16T>A/c.1521_1523delCTT (p.Phe508del, F508del). The study and informed voluntary consent forms were approved by the Ethics Committee of the Ministry of Education and Science of the Russian Federation on 15 October 2018 (the chairman of the Ethics Committee is Professor L. F. Kurilo).

### 2.6. Forskolin-Induced Swelling (FIS) Assay on Intestinal Organoids

When obtaining organoid cultures and performing the FIS assay, the protocols developed under the guidance of J. Beckman were used as a basis [16,17,18]. The method of obtaining intestinal organoids from rectal biopsies at the RCMG was described in detail previously [19]. The patient’s biological material was collected after obtaining informed voluntary consent. Individual crypts were isolated from rectal biopsies and incubated with 10 mM EDTA solution (Thermo Fisher Scientific, Waltham, Massachusetts, MA, USA). The crypts were immersed in a 3D matrix (Matrigel, Corning, New York, NY, USA) and seeded into 24-well plates. After polymerisation of the Matrigel, the growth medium was added. The composition of the medium was described previously [17,19]. Organoids were transplanted once every seven days via the mechanical destruction of large budding structures into small fragments. For the FIS assay, organoids were cultured in 96-well plates. After 24 h, the organoids were stained with Calcein AM (Biotium) and stimulated with forskolin at concentrations of 0.128, 0.8, and 5 µM. The treatment lasted for 60 min. Simultaneously, at certain time points (0, 20, 40, and 60 min), “fixed” fields were captured using an Observer D1 fluorescence microscope (Zeiss, Germany). VX-809 (3.5 µM; Selleckchem, Houston, TX, USA) was added at the organoid seeding stage, and VX-770 (3.5 µM; Selleckchem, Houston, TX, USA) was added simultaneously with forskolin. Quantitative analysis of organoid swelling was performed using ImageJ software. When plotting the graph (Sigma Plot 12.5), the area under the curve of the organoid volume depending on time (AUC, area under the curve) was calculated using Microsoft Excel 2007.

Statistical data processing was performed using STATISTICA 8.0 application software package. Depending on the type of distribution, the central trend and scattering measures were the mean (M) ± standard deviation (SD) or median (Me; interquartile range). To compare categorical variables, we used the Fisher test, and quantitative variables were compared using the Mann-Whitney test. Differences were considered statistically significant at *p* < 0.05.

## 3. Results

According to the Russian Register of Patients with CF in 2018, the allele c.3140-16T>A (3272-16T>A) occurred 20 times and ranked 21st with a frequency of 0.34% [5]. According to the 2018 Registry, 16 of the 21 alleles were identified in patients in the Central Federal District and Volga Federal District. The genetic variant c.3140-16T>A was found in all patients at the first stage of DNA diagnostics, as it is included in the panel of genetic variants of the CFTR gene that are common for the Russian Federation. When sequencing at the second stage, the genetic variant c.3140-16T>A was not detected (according to the 2018 Register, sequencing was performed in 510 patients). The frequency of occurrence of this genetic variant in Europe and other countries has not yet been determined.

The Russian CF patients Registry in 2018 provided information on 19 patients with the genetic variant c.3140-16T>A (3272-16T>A). Among the 19 patients with this genetic variant, nine were children and 10 were adults. Patients with six different genotypes were examined as follows: c.3140-16T>A/c.3140-16T>A, F508del/c.3140-16T>A, 394delTT/c.3140-16T>A, R75X/c.3140-16T>A, 2143delT/c.3140-16T>A, c.3140-16T>A/unknown. The most common genotype was F508del/c.3140-16T>A (13 people, 68.4%).

It was found that the age of patients at the time of registration in the registry in the first group carrying the genetic variant c.3140-16T>A was 2.2-fold higher than that in patients with the genotype F508del/F508del (*p* = 0.007, Table 1). At the same time, in the first group, diagnosis was established later than that in patients in the second group homozygous for the genetic variant F508del (*p* = 0.019). The proportion of adults in the first group was 52.6%, whereas in the second group it was 19.8%. The number of deaths in 2018 in the group of homozygotes for F508del was 1.7%, whereas in the group with the allele c.3140-16T>A, it was 5.2%. According to the results of the sweat test, low values were registered in children with the genetic variant c.3140-16T>A compared to the indicators in patients with the genotype F508del/F508del (*p* = 0.001).

The characteristics of the clinical and instrumental health indicators of the studied patients, separately in the groups of children and adults, are presented in Table 2. According to spirometry indicators (FEV1 and FVC), higher values were obtained in the children in group 1 (*p* = 0.016). However, there were no differences in adults according to the spirometry data. In terms of nutritional status (BMI), there were no significant differences between the groups. The body weight and height indicators of adult patients with the genotype containing the genetic variant c.3140-16T>A were significantly higher than those of patients in the second group (*p* = 0.010 and *p* = 0.037, respectively, Table 2); however, the BMI of the patients in the study groups did not differ.

Table 3 shows the indicators of physical development for children. The children showed significant differences in the percentile and Z criteria of body weight. The weight indicators of children with the genotype c.3140-16T>A significantly exceeded those of children in group 2.

A study of pancreatic function based on the level of pancreatic elastase-1 showed normal levels of the enzyme (more than 200 µg/g of faeces) in the group of patients with the genetic variant c.3140-16T>A in 70% of cases, compared to that in 9.8% in the second group (*p* < 0.001, Table 4). The frequency of microbial pathogens in the respiratory tract (*P. aeruginosa*, *B. cepacia* complex, *S. aureus*, *MRSA*, and nontuberculosis mycobacteria) did not differ between the observation groups (Table 4). The absence of a number of complications, such as meconium ileus, diabetes, pneumothorax, pulmonary haemorrhage, electrolyte disorders (pseudo-Bartter syndrome), amyloidosis and allergic bronchopulmonary aspergillosis, was observed in patients in the first group (Table 4). All of these complications occurred in patients homozygous for F508del (group 2), except for amyloidosis, which was absent.

A comparison of the therapy performed for patients with different genotypes revealed that patients in the first group with the variant c.3140-16T>A received less frequent pancreatic enzymes (*p* < 0.001), ursodeoxycholic acid (*p* < 0.001), and fat-soluble vitamins (*p* = 0.002), which is associated with preserved function of the pancreas, as well as kinesitherapy (*p* = 0.014; Table 5).

Two patients with variant c.3140-16T>A in the genotype were examined using the ICM method; the first patient, homozygous for c.3140-16T>A, whereas the genotype of the second patient was c.3140-16T>A/F508del (Table 6). The clinical and laboratory parameters of the patients corresponded to a variant with a “mild” genotype with preserved pancreatic function (pancreatic elastase > 200 mg/g).

Short-circuit current density (ΔISC) was performed in response to amiloride administration (sodium channel stimulation) in patients with the genetic variant F508del/c.3140-16T>A, and it was determined to be −11.83 ± 3.32 µA/cm^2^. The change in ΔISC in response to the introduction of forskolin (stimulation of the chloride channels) was 11.5 ± 2.35 µA/cm^2^, which reflects the presence of residual function of the chloride channel and occupies an intermediate value between the indicators of healthy individuals and the comparison group, homozygotes for the F508del mutation. In response to the introduction of histamine, ΔISC changes in the negative direction, which reflects the entry of potassium ions into the cells. The current density was 5.58 ± 1.09 µA/cm^2^.

Analysis of the short-circuit current curve in a patient with the genotype F508del/c.3140-16T>A (Figure 1D) indicated that the response to sodium channel stimulation was lower, possibly due to the presence of a class 2 mutation in the genotype F508del, and the function of the chloride channel with forskolin stimulation was similar to that of genotype c.3140-16T>A/c.3140-16T>A (Figure 1C).

When compared with the graph of changes in the current density based on various stimulators in patients with the F508del/F508del genotype (Figure 1B), there was no reaction to the introduction of forskolin, in contrast to the graphs in patients with the c.3140-16T>A variant.

Thus, using the ICM method, it was shown that the chloride channel (CFTR) retained its residual function in the two genotypes, including the genetic variant c.3140-16T>A, and that the genetic variant belongs to the “mild” class, which is associated with these phenotypic manifestations and is not a characteristic of the genetic variant leading to the formation of a stop codon. The consequences of these aberrations lead to a severe disease course. Cultures of intestinal organoids were next obtained from rectal biopsies of two patients with genotypes c.3140-16T>A/c.3140-16T>A and c.3140-16T>A/F508del. In contrast to the healthy control (wt/wt), the studied organoid cultures had reduced lumens and were characterised by irregular shapes, which is a sign of a disruption in the functional activity of the CFTR protein (Figure 2).

The residual function of the CFTR channel and the effect of CFTR modulators (corrector VX-809 and potentiator VX-770) on the restoration of CFTR protein function were studied using the FIS assay. When exposed to forskolin (0.128–5 µM), both organoid cultures responded with swelling compared to the F508del/F508del control (Figure 3 and Figure 4). The response was concentration-dependent (Figure 4). An analysis of the response to 5 µM stimulation with forskolin showed higher conserved CFTR activity for the c.3140-16T>A homozygous genotype (Figure 3 and Figure 4).

Incubation of the c.3140-16T>A/c.3140-16T>A and c.3140-16T>A/F508del organoids with the VX-770 potentiator led to an increase in forskolin-induced swelling. This effect was more pronounced for the heterozygous genotype. The VX-809 corrector enhanced the action of the potentiator but did not have a positive effect on its own (Figure 4). These results differed from those obtained for the control F508del/F508del organoids. In the case of the F508del homozygote, it was shown that each of the modulators separately slightly increased the functionality of the CFTR, and when VX-809 and VX-770 were used together, there was a significant restoration of channel activity.

Bioinformatic analysis of c.3140-16T>A variant by SpliceAI and Human Splicing Finder (HSF) tools revealed that this variant could lead to a cryptic acceptor splice site activation and an extension of exon 20 by 14 nt (delta score SpliceAI was 0.78; consensus value at HSF was 80.17 against 51.23 at wild type) (Figure 5A).

To determine the effect of the intronic variant on *CFTR* pre-mRNA splicing we isolated total RNA from intestinal tissue and nasal epithelium of the patient with a c.3140-16T>A/F508del genotype and her mother (carrier of the F508del variant). Following RT-PCR analysis of *CFTR* mRNA exon 20 structure showed the same PCR-product corresponding to normal *CFTR* transcript for the proband’s and her mother’s cDNA (Figure 5B). Sanger sequencing of the c.3140-16T>A/F508del patient-derived PCR-product revealed the minor signal of the extended exon 20, but its amount was significantly reduced compared to the wild-type isoform. It has been suggested that this may be the result of the nonsense-mediated decay (NMD) of mutant mRNA. To verify this hypothesis, we performed NMD inhibition on the c.3140-16T>A/F508del patient-derived intestinal organoid samples. Sequencing of the obtained PCR-product showed that treatment of the sample with cycloheximide leads to a significant increase in the amount of the transcript from c.3140-16T>A allele (Figure 5C).

Moreover, we performed a minigene splicing assay to confirm that c.3140-16T>A variant leads to a splicing alteration. For this, *CFTR* exon 20 with 300 bp of flanking intronic sequences, containing wild-type or variant c.3140-16T>A, were cloned into pSpl3-Flu2 vector. Obtained plasmids were separately transfected into HEK293T cells and after 48 h total RNA was isolated. RT-PCR analysis revealed that c.3140-16T>A variant leads to *CFTR* exon 20 elongation by 14 nucleotides (Figure 5D,E), resulting in a frameshift and a premature stop codon formation (R1048Nfs*17). This stop codon is located more than 50–55 nucleotides upstream of the last exon-exon junction and probably leads to mRNA degradation by NMD [21]. This is consistent with the results of patient-derived mRNA analysis.

We observed that variant c.3140-16T>A leads to the splicing disruption followed by mRNA degradation through NMD. At the same time, clinical and laboratory data showed that the variant belongs to the “mild” class. We hypothesized that this may be due to the residual expression of the wild-type isoform from the c.3140-16T>A allele. To validate this hypothesis, we analyzed cDNA obtained from the intestinal organoid samples of c.3140-16T>A/c.3140-16T>A homozygous patient and healthy donor before and after NMD inhibition. To identify the minor splicing events, which could not be obtained by routine RT-PCR with Sanger sequencing, we performed a targeted next generation sequencing of PCR-product covering *CFTR* exons 18–22. The analysis of patient-derived sample revealed that in addition to a major mis-spliced isoform with exon 20 elongation, about 6.8% of exon/exon junction (EEJ)-spanning reads mapped to the wild type isoform. Moreover, we observed two minor aberrant isoforms, with in-frame exon 20 skipping (3.5%) and activation of exonic cryptic splice site resulted in an exon 20 shortening by 34 nucleotides (3.3%). In addition, we showed that after NMD inhibition, relative amount of wild type isoform decreased to 2.5%, while the amount of the mutant isoform increased from 86.4% to 92.5%. (Figure 6) Thus we showed that the c.3140-16T>A allele could produce a residual amount of normally spliced transcript resulting in the patient’s mild phenotype. Moreover, we confirmed that the major mis-spliced isoform with 14nt-elongation of exon 20 is partially degraded by NMD.

## 4. Discussion

The pathogenic variant c.3140-16T>A can be attributed to genetic variants in which there is a decrease in the amount of functional protein or its transport to the apical membrane [22]. It is important to study splicing mutations such as c.3140-16T>A, which allows for a better understanding of the splicing process and the mechanism that causes the disease, which can help develop the most relevant therapeutic strategy and thus contribute to an increase in the life expectancy of patients carrying this class of mutations. The phenotypic features of CF patients with the genetic variant c.3140-16T>A were compared with those of patients with the genotype F508del/F508del. CF patients with the allele c.3140-16T>A in their genotype were of higher average age, were older at diagnosis and consisted of a higher proportion of adults. Thus, patients carrying the c.3140-16T>A allele are characterised by better physical development indicators, both in adults and in children, have preserved pancreatic function, as well as the absence of a number of complications, and received enzyme therapy less often than patients with the F508del/F508del genotype. According to the results of clinical and laboratory studies, the phenotype of patients with the genetic variant c.3140-16T>A can be considered “mild”.

According to the results of the ICM method, partial functioning of the chloride channel was observed during forskolin stimulation in the rectal biopsies of patients with the c.3140-16T>A allele carriers. Incubation of the c.3140-16T>A/c.3140-16T>A and c.3140-16T>A/F508del organoids with the VX-770 potentiator led to an increase in forskolin-induced swelling, which indicates a restoration of CFTR function. This effect was more pronounced than that in F508del/F508del homozygotes. At the same time, the corrector VX-809 did not have a positive effect on the function of the chloride channel, but it enhanced the action of the potentiator (Figure 4). These data confirm the results of the clinical analysis of the course of CF in patients with the genetic variant c.3140-16T>A and the response to forskolin in intestinal biopsies during ICM.

In the present study, the c.3140-16T>A variant of the *CFTR* gene was functionally characterised. It was shown that this variant leads to activation of a cryptic acceptor splice site in intron 19, resulting in an extension of exon 20 by 14 nt with frameshift and premature stop codon formation. It is interesting to note that studies conducted on cDNA obtained from the intestinal biopsy of a patient with c.3140-16T>A/F508del showed that the amount of an aberrant transcript was reduced, which is probably the result of NMD. However, when we examined cDNA from the nasal epithelium of the patient with c.3140-16T>A/F508del, we did not observe this imbalance (Appendix A), which suggests that the process of nonsense-mediated mRNA decay is not the same in different tissues.

The effect of the c.3140-16T>A variant on mRNA structure should lead to a loss-of-function effect due to NMD activation and synthesis of a truncated nonfunctional CFTR protein. However, a functional study of the activity of chloride channels showed partial preservation of their functions. This is consistent with clinical and laboratory studies that considered the phenotype of patients with the c.3140-16T>A variant as mild. We suggest that this discrepancy is probably related to the fact that the c.3140-16T>A variant does not lead to a complete splicing alteration and creates a small amount of wild-type transcript, which is sufficient for the partial preservation of CFTR protein function. Deep next-generation sequencing confirmed the presence of small amount (about 7%) of normally spliced transcript in homozygous for the c.3140-16T>A allele patients. It is important to note that such a small amount of wild type transcript cannot be detected by routine methods such as electrophoresis or Sanger sequencing.

A similar mechanism has been previously described for intronic mutations in the *CFTR* gene [23], as well as in other genes [24]. For example, Highsmith et al. [23] showed that the 3849 + 10kbC>T variant leads to aberrant splicing and inclusion of an 84-bp cryptic exon in the *CFTR* mRNA, which contains a stop codon, and its presence leads to a truncated nonfunctional protein. However, the authors showed that this allele also produces a small amount of normally spliced transcripts and leads to a mild patient phenotype.

Thus, the mechanism of pathogenicity of splicing variants is often not obvious and needs to be studied in detail in each specific case. A description of the clinical picture of patients with CF with the genetic pathogenic variant c.3140-16T>A, an assessment of its functional activity using the ICM method, specifically FIS assay on rectal organoids, including the use of CFTR modulators, allows us to confidently attribute it to the genetic variants of class IV-V. The study showed that there was a decrease in the amount of functional proteins. The remaining small amount of wild-type transcript was sufficient to partially preserve the function of the CFTR protein. The milder course of the disease and the prevalence of patients with preserved pancreatic function confirmed the data obtained.

## 5. Conclusions

An integrated approach to the description of a genetic variant, c.3140-16T>A, made it possible to verify a “mild” phenotype in carriers of this allele in the *CFTR* gene and confirm their features with functional tests (method ICM, sweat test) and forskolin-induced swelling assay on rectal organoids. It also facilitated the establishment of a complex mechanism of disorders of the CFTR protein, based on the lack of a complete change in splicing with residual CFTR protein synthesis due to low quantities of normally spliced transcripts. The results of this study could form the basis of the development of targeted therapy for this genetic variant of the *CFTR* gene.

## Figures and Tables

**Figure 1 genes-12-00837-f001:**
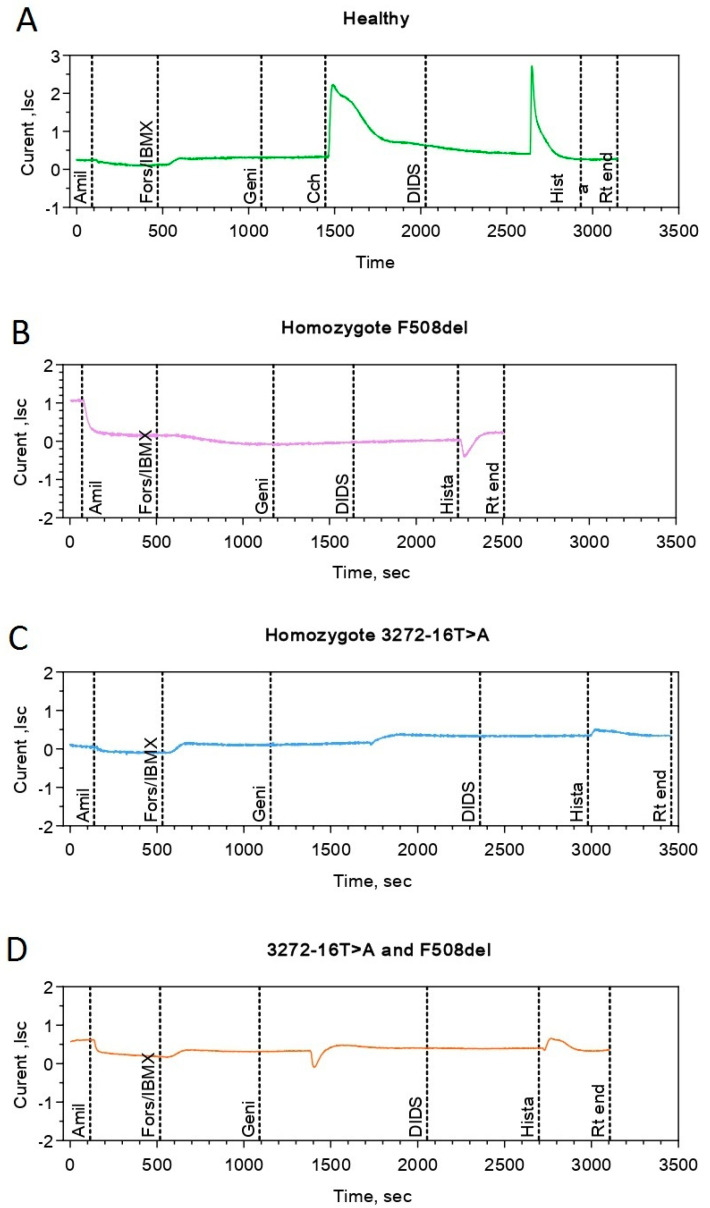
Intestinal current measurement (ICM) for patients carrying the genetic variant c.3140-16T>A. Note: (**A**) Healthy individual (control). The addition of amiloride caused a decrease in ΔISC, there was a significant increase of ΔISC in response to forskolin/IBMX, while the addition of histamine led to a change of short circuit current in the positive direction. (**B**) When amiloride was administered, there was a decrease in the short-circuit current (ΔISC), but no changes were observed in response to the introduction of forskolin/IBMX, and a negative change in the short-circuit current was observed in response to the addition of histamine. (**C**) When amiloride was administered, there was a decrease in the short-circuit current (ΔISC), an increase in the current in response to forskolin/IBMX administration was observed, and a negative change in the short-circuit current was observed with the addition of histamine. (**D**) With the introduction of amiloride, the short-circuit current (ΔISC) decreased. In response to the introduction of forskolin/IBMX, the current increased, and with the addition of histamine, a negative change in the short-circuit current was observed, as in the comparison group.

**Figure 2 genes-12-00837-f002:**
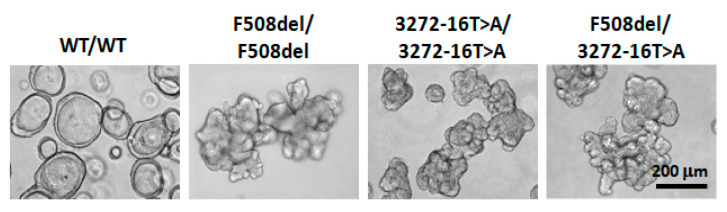
Morphology of intestinal organoids with the genetic variant c.3140-16T>A (3272-16T>A). Organoids obtained from a healthy volunteer (wt/wt) and a patient with the F508del/F508del genotype were used as controls.

**Figure 3 genes-12-00837-f003:**
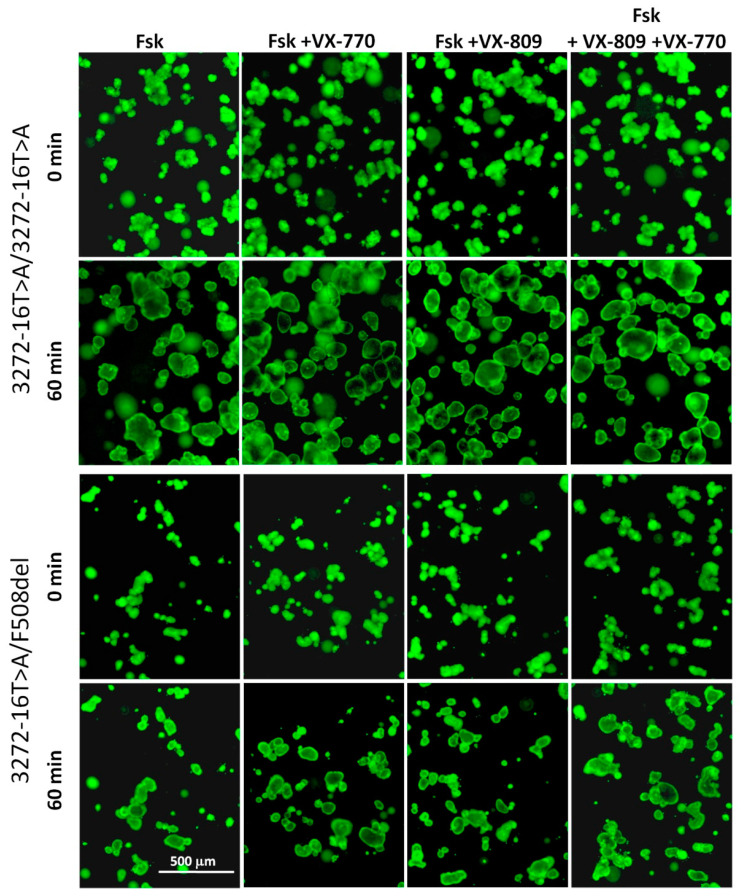
Characteristic images of intestinal organoids with genotypes c.3140-16T>A/c.3140-16T>A and c.3140-16T>A/F508del before exposure to forskolin (fsk; 5 µM) and targeted drugs (both 3.5 µM) and after treatment. Staining-Calcein (0.84 µM, 1 h); objective ×5, scale-500 microns.

**Figure 4 genes-12-00837-f004:**
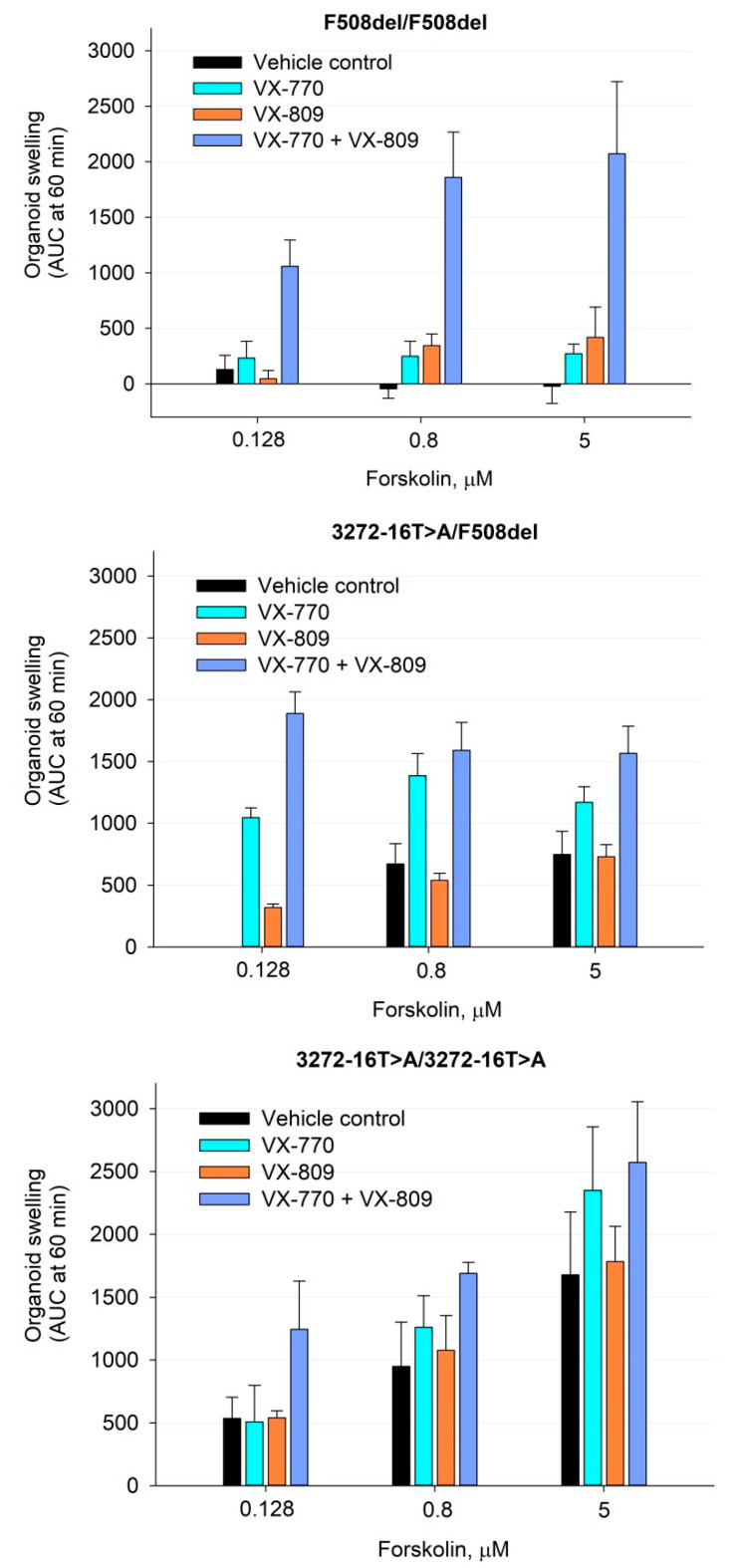
Results of quantitative assessment of organoid swelling under the action of forskolin and CFTR modulators for patients with the genetic variant c.3140-16T>A (3272-16T>A) and F508del/F508del control. Working concentrations-3.5 µM for VX-770 and 3.5 µM for VX-809. AUC, area under the curve.

**Figure 5 genes-12-00837-f005:**
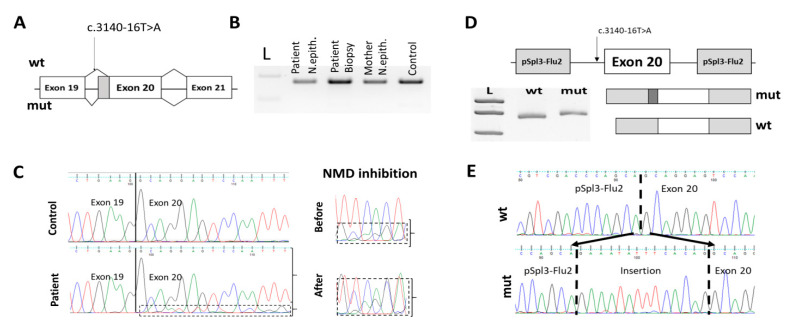
RT-PCR and minigene analysis of c.3140-16T>A. (**A**) Scheme of the locus containing c.3140-16T>A and expected splicing products. (**B**) Electrophoregram of RT-PCR products. (**C**) Sequencing of RT-PCR products. The dashed line outlines the mutant isoform of exon 20. (**D**) Scheme of the minigene construct and PAGE of RT-PCR fragments obtained from the wild type and mutant minigene. (**E**) Sequencing of minigene products.

**Figure 6 genes-12-00837-f006:**
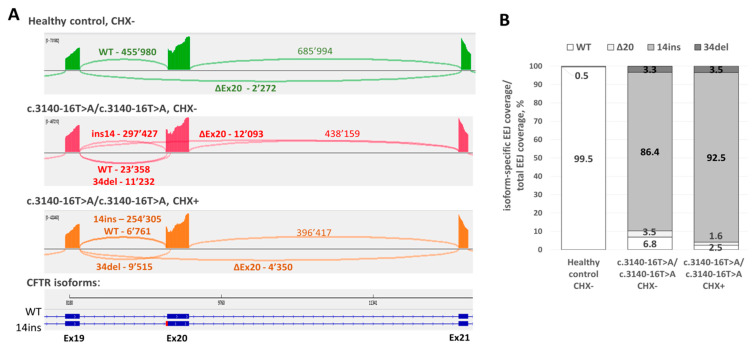
Targeted next generation sequencing of PCR-products covering *CFTR* exons 18–22 and obtained from intestinal organoid samples of healthy control and c.3140-16T>A homozygous patient before (CHX-) and after (CHX+) NMD inhibition by 300 μM cycloheximide treatment. (**A**) Sashimi plots to visualize splice junctions in different samples. Numbers represent the number of reads spanning indicated exon/exon junction (EEJ). (**B**) The relative amount of each *CFTR* isoforms in samples are represented as percentage of isoform-specific EEJ-coverage from total EEJ coverage.

**Table 1 genes-12-00837-t001:** General characteristics of patients with the studied genotypes.

Indicator	c.3140-16T>A/Other	F508del/F508del	*p*
	(1 Group)	(2 Group)	
Average age *, years			0.007
Me (range)	21.4 (8.8–31.3)	9.7 (5.2–15.6)	
Mean (SD)	20.3 ± 13.6	11.5 ± 8.4	
N	19	881	
Age of diagnosis, years			0.019
Me (range)	0.3 (0.2–13.2)	0.3 (0.1–1.3)	
Mean (SD)	7.4 ± 10.2	1.5 ± 3.2	
N	19	870	
Sweat test (conductivity), mmol/l			0.001
Me (range)	100.0 (93.0–105.0)	112.0 (102.0–121.5)	
Mean (SD)	98.0 ± 9.4	111.4 ± 16.07	
N	11	332	
Adults, n, %	10 (52.6)	174 (19.8)	0.002
Deceased, n, %	1 (5.2)	15 (1.7)	-

Note: * average age of patients at the time of registration.

**Table 2 genes-12-00837-t002:** Characteristics of children and adults with cystic fibrosis with different genotypes.

Indicator	Adults	Adults	*p*	Children	Children	*p*
	c.3140-16T>A/Other	F508del/F508del		c.3140-16T>A/Other	F508del/F508del	
BMI			0.092			0.106
Me (range)	20.4 (18.4–24.1)	18.8 (17.4–20.2)		16.3 (15.7–17.9) 16.6 ± 1.6	15.4 (14.3–16.7)	
Mean (SD)	21.8 ± 5.9	18.9 ± 2.4		8	15.7 ± 2.1	
N	10	170			693	
FEV_1,%_			0.401			0.016
Me (range)	60.0 (28.0–69.8)	61.2 (42.0–80.0)		115.5 (100.0–123.7)	87.0 (72.0–100.0)	
Mean (SD)	53.5 ± 29.2	62.7 ± 27.0		111.8 ± 16.6	85.3 ± 21.2	
N	7	134		4	378	
FVC,%			0.653			0.043
Me (range)	73.0 (56.0–96.9)	80.0 (62.0–96.1)		108.0 (99.5–120.6)	91.0 (78.0–102.0)	
Mean (SD)	74.4 ± 25.1	79.8 ± 23.4		110.1 ± 17.4	89.2 ± 19.1	
N	7	133		4	375	

**Table 3 genes-12-00837-t003:** Indicators of body weight and height (percentiles and Z-criteria) in patients with cystic fibrosis with different genotypes in childhood.

Indicator	c.3140-16T>A/Other	F508del/F508del	*p*
	(1 Group)	(2 Group)	
Weight, percentiles			0.014
Me (range)	70.0 (42.1–97.1)	31.2 (11.1–59.1)	
Mean (SD)	69.5 ± 28.3	36.9 ± 29.1	
N	6	446	
Weight, Z-criteria Me (range)	0.7 (−0.2–1.9)	−0.5 (−1.2–0.2)	0.014
Mean (SD)	0.8 ± 1.0	−0.5 ± 1.1	
N	6	446	
Height, percentiles			0.215
Me (range)	44.4 (25.3–88.0)	34.1 (10.7–63.1)	
Mean (SD)	52.9 ± 35.1	39.0 ± 30.9	
N obc.	8	656	
Height, Z-criteria			0.215
Me (range)	−0.1 (−0.7–1.3)	−0.4 (−1.2–0.3)	
Mean (SD)	0.2 ± 1.3	−0.4 ± 1.2	
N	8	656	

**Table 4 genes-12-00837-t004:** Clinical characteristics of cystic fibrosis patients with different genotypes.

Clinical Features			c.3140-16T>A/Other (1 Group)		F508del/F508del (2 Group)		*p*
			n	%	n	%	
pancreatic elastase-1 *	PS (>200 mcg/g)		7	70	44	9.8	<0.001
	PI (<200 mcg/g)		3	30	406	90.2	
microorganisms	*P. aeruginosa*	Chronic **	6	33.3	295	33.9	>0.99
		Intermittent	3	17.6	133	15.7	0.740
	*S. aureus*	Chronic **	9	52.9	526	60.8	0.618
	*MRSA*		2	11.8	30	3.5	0.126
	*B. cepacia* *complex*	Chronic **	2	11.8	59	6.8	0.330
	Nontuberculosis mycobacteria		1	7.1	5	0.7	0.106
	*S. maltophilia*		1	5.9	32	3.7	0.482
	*Achromobacter* spp		0	0	48	5.6	0.618
	*NFGNF*		0	0	98	11.1	0.246
	*Haemophilus influenzae*		2	12.5	52	6.2	0.267
Osteoporosis			3	25.0	46	7.7	0.065
Polyposis			6	31.6	253	30.7	>0.99
Meconium ileus			0	0	89	10.2	-
ABPA			0	0	14	1.6	-
Diabetes (treatment with insulin daily)			0	0	30	3.4	-
Pneumothorax			0	0	9	1.0	-
Pulmonary hemorrhage			0	0	8	0.9	-
Oncological disease			0	0	1	0.1	-
Pseudo-Bartter syndrome			0	0	25	2.9	-
Amyloidosis			0	0	0	0	-
Liver transplantation			0	0	3	0.4	-
Lung transplantation			1	5.3	11	1.2	0.227

Note: * the calculation was carried out according to the data presented in the register; **chronic infection according to the criteria proposed by Lee et al. (2003) [20], according to which the detection of the pathogen in more than 50% of sputum samples or flushes during the previous 12 months can be interpreted as a chronic infection.

**Table 5 genes-12-00837-t005:** Therapy for patients with cystic fibrosis with different genotypes.

	c.3140-16T>A/Other 1 Group)		F508del/F508del (2 Group)		*p*
	n	%	n	%	
Inhalation of hypertonic solution	11	50.9	627	72.0	0.199
Inhaled antibiotics	7	36.8	417	47.7	0.487
Intravenous antibiotics	7	36.8	342	39.3	>0.99
Oral antibiotic	16	84.2	540	61.8	0.055
Bronchodilators	13	68.4	444	50.8	0.165
Inhaled steroids	4	21.1	134	15.3	0.517
Oral steroids	1	5.3	39	4.5	0.585
Oxygen therapy	2	10.5	28	3.2	0.131
Dornase Alfa	18	94.7	852	97.1	0.432
Azithromycin	4	21.1	287	33.2	0.330
Ursodeoxycholic acid	11	57.9	820	93.5	<0.001
Pancreatic enzymes	10	52.6	869	99.2	<0.001
Proton Pump inhibitors	3	15.8	175	20.1	0.779
Fat-soluble vitamins	14	73.7	827	95.1	0.002
Kinesiotherapy	12	63.2	741	85.7	0.014

**Table 6 genes-12-00837-t006:** Indicators of the short-circuit current density (ΔISC) during the administration of stimulants to patients carrying the genetic variant c.3140-16T>A.

	ΔISC, µA/cm^2^	Amiloride	Forskolin/IBMX	Genistein	DIDS	Histamine
Homozygous 3272-16T>A	Biopsy № 1	−9	15	2	1.5	9.5
	Biopsy № 2	−2.5	3.5	2	1.5	4.5
	Biopsy № 3	−6.5	16.5	2	1.5	7
	M ± m patient 1	−6 ± 2.32	11.67 ± 5.03	2	1.5	7 ± 1.77
Heterozygous c.3140-16T>A/F508del	Biopsy № 1	−19.5	10	2	2	3.5
	Biopsy № 2	−12	6	2	2	3
	Biopsy № 3	−21.5	18	2	2	6
	M ± m patient 2	−17.67 ± 3.54	11.33 ± 4.32	2	2	4.17 ± 1.14
M ± m common Me		−11.83 ± 3.32 10.5	11.5 ± 2.35 12.5	2	1.75 ± 0.12 1.75	5.58 ± 1.09 5.25
F508del/F508del		−18.39 ± 5.62	3.06 ± 0.89	1.83 ± 0.35	1.83 ± 0.35	21.5 ± 5.46
Healthy individuals		−8.98 ± 3.42	25.78 ± 4.41	2 ± 0.29	1.8 ± 0.26	101.68 ± 10.99

## Data Availability

The datasets used and/or analysed during the current study are available from the corresponding author upon reasonable request.

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
