# Peer review of "Health Characteristics of Patients with Cystic Fibrosis whose Genotype Includes a Variant of the Nucleotide Sequence c.3140-16T>A and Functional Analysis of this Variant"

_genes, 2021, doi:10.3390/genes12060837_

Round 1
Reviewer 1 Report
The clinical characteristic of patients with rare CF mutations is extremely important to increase our knowledge about the disease as well as provide the patients with prognostic information. It also might be beneficial in the context of developing management strategies for the less frequent mutations. The authors are describing the CF patients with c.3140-16T>A variant from Russian CF Registry and provide some data on its functional analysis.
There are however some minor issues that authors need to address
- The main concern would be if you could consider “mild” a mutation that might lead to premature death (5.2% of patients) and leads to the lung function impairment - FEV1 in adults is actually worse than in adults homozygous for F508del. Pancreatic function is not the only indicator of the disease severity
- Introduction: verse 59 what does “frequently occurring” CF variants mean? There are 23 CFTR pathogenic mutations with the incidence >0.5%. What is the incidence of these 99 mentioned by the authors?
- Materials and methods: it would be interesting to evaluate the influence of the second mutation in patients with c.3140-16T>A on the clinical course of the disease
- Materials and methods: verse 132 why did the authors consider 26th-75th percentile normal and not 5th-95th ?
- Materials and methods verse 209-210 – what was the second step in the procedure -the authors describe the first and the third step?
- Table 2 – mean and median of the raw values of weight and height in paediatric patients are unnecessary and irrelevant – they are age dependent so z-scores and percentiles are the appropriate way to present it (as in Table 3)
- Results verse 320 authors state that amyloidosis was present in the patients homozygous for F508del. Yet in the table 4 the number of patients with amyloidosis is 0. Which is correct?
- Discussion – does forskolin-induced swelling indeed indicate a disruption of channel function or just the opposite ?– verse 512
- Figure 1 – there is no description of graph A
- What is formalin test – verse 399, preformed antibiotics – table 5 or kinesitherapy – table 5?; What does “MB” stand for – verse 500
- English proof-reading and editing is strongly recommended; eg: chloride and chlorine channel are used interchangeably.
Author Response
Response 1: Рreservation of pancreatic function according to the classification of genetic variants of CFTR allows it to be classified as “mild”, one of the characteristics of which is late diagnosis, lower sweat test rates, and better lung function indicators.
Response 2: In the register of patients with CF of the Russian Federation in 2018, 210 pathogenic variants of CFTR are listed, of which 99 occurred more than 1 time [5] 36 pathogenic CFTR mutations occurred with a frequency of> 0.1%.
Response 3: Рatients had different genetic variants in the heterozygous state with mutation c.3140-16T> A, therefore, it was not possible to assess the clinical course of the disease due to the small number of patients in the groups
Response 4: These criteria are consistent with Castellani, C .; Duff, A. J. A .; Bell, S. C .; Heijerman, H. G. M .; et al. ECFS best practice guidelines: the 2018 revision. J. Cyst. Fibros. 2018, 17 (2), 153-178.
Response 5: line 205, highlighted in the text
Response 6: corrected, removed
Response 7: Amyloidosis was absent, explanations were made in the text.
Response 8: The reviewer was absolutely right, not a disruption, but a restoration CFTR function! Error, corrected.
Response 9: Inserted
Response 10: Errors, corrected
Response 11: The chloride channel is indicated everywhere
We attach an article with edits.
Reviewer 2 Report
Kondratyeva et al. present a well-rounded, multi-modal study that provides insights into c.3140-16T>A variant of CFTR. The authors precisely characterize the phenotype, which is mild due to the randomness of splicing (3% of transcripts were correct). Although the variant is rare and most of the CFTR mutations have been studied well, this story is exciting, compelling, and fresh.
Only minor remarks:
Line 17 – remove “in” ?
Line 21 (abstract) – please specify that “enzymes” refers to pancreatic enzyme replacement therapy (dornase-alpha is also an enzyme). I would suggest “and required pancreatic enzyme replacement therapy less often…”
Line 77 – “this” – “this variant”?
Line 91 – 81 regions. How many regions are there in total in the country? Is 81 all the regions or a fraction of all the regions?
Lines 220-227 “were used as a control for the obtained data” – is unclear to me; in the end it is difficult to understand how many patients were involved (1 with the investigated mutation), the number of patients with F508del seems to be reported twice. But then in lines 337-339 it becomes clear that there were two patients carrying c.3140-16T>A: one was homozygous, and other compound heterozygous (with the other allele being F508del).
Lines 259-263 What is the proportion of Registry entries containing information from exome sequencing? Is this region well-covered by the employed exome screening technique? (remark of secondary importance)
L271-L275 Some more context is needed – the reader may wonder what is the newborn screening strategy in Russia.
Table 2. The comparisons between the two groups are difficult because of inherent differences. The authors did their best to present these data clearly. Does “Me” is intended to stand for “mean”?
Was the c.3140-16T>A patient who received the lung transplantation the one who died?
“Preformed” – do the authors mean “Oral antibiotics” “Oral steroids”?
L339 “formalin test” – forskolin?
L401 “Ann analysis” – “An analysis”
L461 “wich” – “which”
L416 “couldn’t” – “could not”
L467 “activtion” – “activation”
L471 “resulted” – “resulting”
Fig. 5A Please improve figure resolution, if possible.
L500 “MB” unclear
L563 “low quantities normally splicing transcripts” please consider “low quantities of normally spliced transcripts”
The conclusion is “The results of this study could form the basis of the development of targeted therapy for this genetic variant of the CFTR gene.” – I partially agree, I think this conclusion can be left as is. But there are too few patients and the development of targeted treatment is very laborious. I think that this study demonstrates the high relevance of in-depth clinical and molecular investigation for the understanding of surprising phenotypes. If monogenic disease is so complex and demands so much work to understand, how will we deal with problems that involve multifaceted gene-environment interactions? It will be even harder.
Author Response
Response 1: error, corrected
Response 2: made adjustments
Response 3: error, corrected
Response 4: There are 85 regions in the Russian Federation. The register contains data from 81 regions. 4 regions did not report data in 2018.
Response 5: The data are indicated by us correctly for patients with the genetic variant c.3140-16T>A
Response 6: According to the DNA diagnostic algorithm in the Russian Federation, the genetic variant c.3140-16T> A is included in the analysis fo common variants at the 1st stage of the study and was found in all patients. When sequencing at the 2nd stage, this genetic variant was not detected. When sequencing according to Sanger, as well as when using NGS technologies, the region of the c.3140-16T>A variant is included in the study area, and the coverage at this point is sufficient (always greater than x10). According to the 2018 Registry, 510 patients were sequenced.
Response 7: Variant 3272-16TA - out of 19 patients - 9 (47%) were diagnosed by neonatal screening.
Variant F508del - out of 881 patients - 467 (53%) were diagnosed by neonatal screening.
Response 8: Me - median
Response 9: The deceased patient with mutation 3272-16T>A did not have transplantation
Response 10: error, corrected
Response 11: error, corrected
Response 12: error, corrected
Response 13: error, corrected
Response 14: corrected
Response 15: error, corrected
Response 16: error, corrected
Response 17: Unfortunately, this is not possible.
Response 18: error, corrected
Response 19: error, corrected
We attach an article with edits.